# Antibiotic Utilization in Acute Pancreatitis: A Narrative Review

**DOI:** 10.3390/antibiotics12071120

**Published:** 2023-06-28

**Authors:** Andrea Severino, Simone Varca, Carlo Airola, Teresa Mezza, Antonio Gasbarrini, Francesco Franceschi, Marcello Candelli, Enrico Celestino Nista

**Affiliations:** 1Digestive Disease Center, Fondazione Policlinico Universitario Agostino Gemelli IRCCS, 00168 Rome, Italy; andrea.severino01@icatt.it (A.S.); simone.varca01@icatt.it (S.V.); carlo.airola01@icatt.it (C.A.); teresa.mezza@policlinicogemelli.it (T.M.); antonio.gasbarrini@unicatt.it (A.G.); 2Dipartimento Universitario di Medicina e Chirurgia Traslazionale, Università Cattolica del Sacro Cuore, 00168 Rome, Italy; 3Department of Emergency, Anesthesiological and Reanimation Sciences, Università Cattolica del Sacro Cuore, Fondazione Policlinico Universitario A. Gemelli IRCSS, 00168 Rome, Italy; francesco.franceschi@policlinicogemelli.it (F.F.); marcello.candelli@policlinicogemelli.it (M.C.)

**Keywords:** antibiotics, prophylaxis, acute pancreatitis, probiotics

## Abstract

Acute pancreatitis is a complex inflammatory disease with significant morbidity and mortality. Despite advances in its management, the role of antibiotics in the prophylaxis and treatment of acute pancreatitis remains controversial. The aim of this comprehensive review is to analyze current evidence on the use of antibiotics in acute pancreatitis, focusing on prophylactic and therapeutic strategies. Prophylactic use aims to prevent local and systemic infections. However, recent studies have questioned the routine use of antibiotics for prophylaxis and highlighted the potential risks of antibiotic resistance and adverse effects. In selected high-risk cases, such as infected necrotizing pancreatitis, prophylactic antibiotic therapy may still be beneficial. As for therapeutic use, antibiotics are usually used to treat infected pancreatic necrosis and extrapancreatic infections. When selecting an antibiotic, the microbiologic profile and local resistance patterns should be considered. Combination therapy with broad-spectrum antibiotics is often recommended to cover both Gram-positive and Gram-negative pathogens. Recent research has highlighted the importance of individualized approaches to antibiotic use in acute pancreatitis and underscored the need for a tailored approach based on patient-specific factors. This review also highlights the potential role of new antimicrobial agents and alternative strategies, such as probiotics, in the management of acute pancreatitis.

## 1. Introduction

Acute pancreatitis (AP) is a serious inflammatory process that can range from a mild, self-limiting condition to a life-threatening disease with significant morbidity and mortality. The exact mechanisms that trigger the onset of AP are not yet fully understood but are thought to involve a complex interaction of genetic, environmental, and lifestyle factors. The initial event in the development of AP is the inappropriate activation of pancreatic enzymes within the acinar cells; under pathological conditions, such as bile reflux, alcohol abuse, or gallstones, the activation of pancreatic enzymes occurs prematurely within the pancreas itself. Once activated, they can directly damage acinar cells, leading to their necrosis. The release of pro-inflammatory cytokines, such as interleukin-1β (IL-1β) and tumor necrosis factor-alpha (TNF-α), further amplifies the inflammatory response. These cytokines recruit and activate immune cells, including neutrophils and macrophages, which infiltrate the pancreas and release additional inflammatory mediators, exacerbating tissue damage and promoting the production of reactive oxygen species (ROS), resulting in oxidative stress [1]. Calcium signaling also plays a critical role: intracellular calcium overload occurs due to both the release from intracellular stores and the influx from the extracellular space. Elevated cytosolic calcium levels impair cellular functions and promote the activation of destructive enzymes. Calcium also acts as a trigger for the activation of various signaling pathways involved in inflammation and cell death [2].

Another crucial pathophysiological mechanism underlying AP is bacterial translocation, which refers to the migration of bacteria and their products from the gastrointestinal tract to extraintestinal sites; possible disruption of the intestinal barrier allows gut bacteria to migrate into the systemic circulation and reach the pancreas, triggering local and systemic inflammatory responses [3].

The Atlanta criteria, first established in 1992 and revised in 2012, are widely used to standardize the diagnosis and classification of AP [4]. According to these criteria, AP is diagnosed when two of the following three conditions are met: acute onset of persistent, severe epigastric pain radiating to the back; serum amylase and/or lipase levels greater than three times the upper normal limit; and imaging findings suggestive of acute pancreatitis on ultrasound, computed tomography (CT), or magnetic resonance imaging (MRI). In terms of severity, the disease is classified as mild, moderate, or severe depending on the presence of systemic complications, such as organ failure and necrosis. Mild AP has no organ failure or local or systemic complications; moderate AP has transient organ failure, local complications, or exacerbation of comorbid conditions; and severe AP is defined by the presence of persistent organ failure [4].

In addition to the Revised Atlanta Classification, several scoring systems have been developed to assess the severity and prognosis of AP: the Ranson criteria, which evaluate clinical and laboratory parameters at admission and within 48 h to predict the severity of the disease [5]; the Acute Physiology and Chronic Health Evaluation (APACHE) II score [6]; and the Bedside Index for Severity in Acute Pancreatitis (BISAP) [7]. The latter two incorporate additional factors to provide a more comprehensive assessment of disease severity.

Advancements in imaging techniques have also contributed to classification. CT scans are routinely used to assess the extent of pancreatic inflammation and detect complications such as necrosis, abscess formation, and pseudocysts. The Balthazar scoring system, based on CT findings, categorizes acute pancreatitis into several grades ranging from A to E, depending on the severity and extent of pancreatic involvement [8].

In this heterogeneous clinical setting, the indications and timing for antibiotic therapy have been the main controversial topics in recent decades. The results of the first meta-analyses indicated a beneficial effect of antibiotic prophylaxis (with a significant reduction in mortality [9,10,11,12], pancreatic infections [11,12], and sepsis [10,12]) in patients with severe AP (SAP). The start of big randomized, double-blind, placebo-controlled trials on the other hand, i.e., Refs. [13,14,15,16,17], has not confirmed the previously recognized benefit of antibiotic therapy. Renewed evidence from randomized clinical trials (RCTs) has been confirmed by recent systematic reviews [18,19,20,21,22], and antimicrobials are currently recommended only when infection is either proven or strongly suspected [23]. The aim of this review is to analyze the available literature on the use of antibiotics in the prophylaxis and/or treatment of AP.

## 2. Prophylactic Antibiotics

### 2.1. Guidelines and Latest Evidences

Prophylaxis is the administration of antibiotics to patients who do not have a clinical infection with the goal of preventing pancreatic infection [24]. According to the most recent published guidelines [24,25,26,27], the use of prophylactic antibiotics is not recommended in patients with predicted SAP and necrotizing AP because their use is not associated with a significant reduction in mortality or morbidity. Although this recommendation refers specifically to patients with SAP, it should be made clear that prophylactic antibiotics have no place in the treatment of less severe forms of AP [27]. Since the publication of the last guidelines, several meta-analyses have been performed to investigate the role of antibiotic prophylaxis in this setting and to support these recommendations. In 2020, Ding et al. conducted a meta-analysis [21] that included 11 RCTs and 747 participants divided into two groups: the first one received prophylactic antibiotic therapy (*n* = 376), while the second group did not receive antibiotic therapy and was considered the control group (*n* = 371). The study did not identify significant differences in the incidence of infected pancreatic necrosis, surgical intervention, and mortality when comparing the two groups. Interestingly, antibiotic prophylaxis was associated with a statistically significant decrease in the incidence of nonpancreatic infections. A subsequent meta-analysis [23] (involving 21 studies and 1383 patients) confirmed these findings and showed some beneficial effects of antibiotic prophylaxis (reduction in length of hospital stay, overall infection rates, and nonpancreatic infections), but without significant effects on mortality; in particular, the risk of urinary tract infections and sepsis was significantly reduced by antibiotic prophylaxis. Guo et al. conducted another meta-analysis [28] in 2022 to determine the efficacy of prophylactic carbapenem antibiotics in SAP (7 articles, including 5 randomized controlled trials and 2 retrospective observational studies with an average of 3.864 participants). The study showed a statistically significant decrease in the incidence of infections and complications between the intervention and control groups, but subgroup analysis showed no statistically significant differences. No difference was found in the incidence of infected pancreatic or peripancreatic necrosis mortality; nonpancreatic infections; pulmonary, blood, and urinary tract infections; pancreatic pseudocysts; fluid collections; organ failure; acute respiratory distress syndrome; need for surgery; dialysis; need for invasive or noninvasive ventilation; and intensive care). In summary, recent studies confirm that antibiotic prophylaxis in the setting of SAP is not associated with a statistically significant reduction in mortality or incidence of infected pancreatic necrosis (IPN), with variable reductions in the incidence of nonpancreatic infections. Table 1 provides an overview of meta-analyses performed in recent decades and their results in terms of reduction in mortality, incidence of IPN, and nonpancreatic infections. The differences in results are likely due to differences in the class of antibiotics administered, timing of therapy, treatment outcomes, and design of the studies included in the analysis.

### 2.2. Timing of Administration

In 2008, Xu and Cai [29] stated that the timing of antibiotic prophylaxis is crucial. Early prophylactic antibiotic administration could reduce the incidence of IPN. A 2015 meta-analysis [30] addressed this question. The analysis included 6 RCTs in which patients were treated with antibiotics within 72 h of symptom onset or 48 h of hospital admission (with a total of 397 patients included). The results showed that the mortality rate and incidence of IPN were significantly reduced in patients who received antibiotics, confirming the original hypothesis that the timing of administration is critical.

### 2.3. Role of Procalcitonin and PCR in Predicting Infected Pancreatic Necrosis

Several studies suggest that patients with high levels of C-reactive protein (CRP) [31,32] and procalcitonin (PCT) [33,34,35,36] are at increased risk for IPN [37]. PCT serum measurements can be useful in predicting the likelihood of developing infected pancreatic necrosis [24] and can help identify patients who might benefit from early antibiotic therapy. Algorithms based on PCT measurements have been shown to be effective in distinguishing bacterial sepsis from systemic inflammatory response syndrome (SIRS) in a variety of settings [38]. In 2009, Mofidi et al. [39] found that the sensitivity and specificity of PCT serum levels for predicting IPN were 0.80 and 0.91, respectively, and suggested that serial serum measurements of PCT may be valuable for predicting the severity of AP and the risk of developing IPN. In 2002, a randomized controlled trial [40] was conducted to investigate whether the initiation, continuation, and termination of antibiotic treatment depending on a procalcitonin-based algorithm could result in lower antibiotic use without having a negative impact on the outcome of acute pancreatitis. The results of this study suggest that serum procalcitonin levels may be useful in reducing antibiotic use in patients with AP without causing more infections or harm. Similar to PCT, serial measurements of CRP during the course of SAP are recommended to assess disease severity [37,41]. In 2017, Stirling et al. [42] demonstrated that an interval change in CRP is a comparable measure to absolute CRP levels in predicting the severity of AP. More specifically, they found that an increase of at least 90 mg/dL since admission or an absolute value of 190 mg/dL after 48 h predicted the severe form of AP with the greatest accuracy.

### 2.4. Adverse Events of Antibiotic Prophylaxis

Despite its possible clinical benefits, antibiotic prophylaxis is also correlated with risks, including increased prevalence of multi-drug-resistant bacteria and incidence of fungal infection, among other effects [21,28]. Fungal infection is a serious consequence of AP that is related to an increase in morbidity and mortality [43]. In May 2009, Xue et al. conducted an RCT in order to assess whether antibiotic prophylaxis (imipenem and cilastatin) could be beneficial for acute necrotizing pancreatitis; the study found no benefit in terms of mortality or reduction in infected pancreatic necrosis, but interestingly it found an Increased incidence of fungal infection in patients treated with antibiotic prophylaxis when compared to placebo [44]. Another retrospective cohort study conducted in 2019 by Horibe et al. found that antimicrobial prophylaxis was significantly associated with the development of invasive pancreatic candidiasis [45]. However, results from the latest metanalysis [21,23] did not show a statistically significative increase in fungal infection incidence in patients treated with antibiotic prophylaxis: Ding et al. found that 12.2% of patients who were administered antibiotic prophylaxis developed a fungal infection compared to 17.7% in the control group (OR, 0.95; 95% CI, 0.30–3.03); Poropat et al. found similar results (16/260 vs. 20/274; RR 0.84; 95% CI 0.43 to 1.61). More research is needed to determine the risk of fungal infections as an adverse event of antibiotic prophylaxis. To date, routine antifungal prophylaxis is not recommended for preventing fungal infections in patients with severe AP [26], regardless of the use of antibiotic prophylaxis.

Although antibiotic prophylaxis is not recommended by international guidelines, the clinical use of prophylactic antibiotics in SAP has become widespread; a retrospective study conducted in the United Kingdom in 2018 [46] showed how the rate of antibiotic use in AP was about 62%, while a global systematic overview of reports that included data on the use of antibiotics showed a use rate of 41–88%. Based on these data, greater adherence to guidelines should be recommended in order to limit the effects of the side effects just described in the face of benefits that are not evident to date based on the most recent evidence.

## 3. Infected Pancreatitis and Antibiotic Therapy

Several guidelines clearly recommend antibiotics for the treatment of infected severe acute pancreatitis [24,25]. An important criterion for the use of antibiotics is the presence of IPN. It occurs in approximately 30% of patients with SAP and is associated with a significantly higher risk of death. Therefore, early detection and timely intervention with antibiotics are critical to improving patient outcomes. This condition typically presents with persistent fever, abdominal pain, and systemic signs of sepsis. The diagnosis of IPN is usually confirmed by imaging studies such as CT or MRI, which demonstrate the presence of gas bubbles or fluid collection within the pancreas. Another condition that can occur in AP is infected peripancreatic fluid collections (IPFCs). IPFCs are sterile fluid collections that develop around the pancreas in response to the inflammatory process. However, in some cases, these collections can become infected with bacteria, leading to abscess formation. Infected IPFCs often present with persistent abdominal pain, localized tenderness, and signs of systemic inflammation. CT or MRI scans are also useful in diagnosing infected IPFCs, as they can show the presence of gas or thickened walls within the fluid collections [47].

Historically, surgical intervention, such as open necrosectomy, was the primary treatment modality of IPN [48]. However, this approach was associated with high morbidity and mortality rates. As a result, less invasive techniques have been explored, leading to the development of the step-up approach, which entails a sequential treatment algorithm that starts with conservative measures and gradually progresses to more invasive interventions if necessary [49]. The treatment begins with initial medical management, including fluid resuscitation, pain control, and broad-spectrum antibiotics, to stabilize the patient and control systemic complications. Serial imaging is crucial for monitoring disease progression and identifying potential complications. If the patient fails to respond to conservative therapy or develops complications, the next step in the approach is minimally invasive intervention: percutaneous catheter drainage (PCD) under radiological guidance. It allows for direct access to the necrotic collection, facilitating drainage and potentially reducing the risk of infection spread. The drainage catheters can be placed either transgastrically or transcutaneously, depending on the location of the necrotic tissue. If the patient demonstrates a favorable response to PCD, further intervention may not be necessary. However, if there is persistent infection or inadequate drainage, the next step involves endoscopic intervention. Endoscopic necrosectomy is a minimally invasive technique that allows for direct visualization and debridement of the necrotic tissue. This approach utilizes endoscopic ultrasound guidance to identify the necrotic areas and deliver therapeutic interventions, such as tissue debridement and cystogastrostomy [50]. In cases where endoscopic intervention fails or is not feasible, the final step in the step-up approach is surgical intervention. This may involve open necrosectomy or minimally invasive surgical techniques, such as video-assisted retroperitoneal debridement or laparoscopic approaches. Surgical intervention is typically reserved for patients with extensive necrosis, multiple collections, or those who fail to respond to less invasive measures [48].

The American Gastroenterological Association Institute guidelines state that carbapenems, quinolones, and metronidazole, which are known to penetrate pancreatic necrosis, may be helpful in individuals with infected necrosis and may be useful to delay or sometimes avoid surgical intervention and reduce morbidity and mortality [25]. The optimal duration of antibiotic therapy for infected AP depends on several factors, including the extent of infection, clinical response, and radiologic findings. In general, a minimum treatment duration of 14 days is recommended. However, in the absence of clear evidence, the decision to extend or discontinue antibiotics should be made on an individual basis, taking into account the patient’s clinical course and the resolution of the infection [25]. It should be noted that the use of antibiotics in AP should be part of a comprehensive treatment strategy that includes early restoration of hydration, pain control, nutritional support, and consideration of invasive measures such as drainage of infected collections or necrosectomy, if necessary. The spectrum of empiric antibiotic therapy should include both aerobic and anaerobic Gram-negative and Gram-positive microorganisms [24]. Fluoroquinolones and carbapenemics, because of their ability to penetrate the pancreatic parenchyma, are generally recommended for confirmed IPN [28]. Aminoglycoside antibiotics in standard intravenous doses have low penetration ability into pancreatic tissue, so they are not the first choice for IPN [51]. Third-generation acylureidopenicillins and cephalosporins have intermediate penetration capacity into pancreatic tissue and are effective against Gram-negative microorganisms. Only piperacillin/tazobactam has shown efficacy against Gram-positive bacteria and anaerobes [52]. Quinolones and carbapenems both show good tissue penetration into the pancreas and excellent anaerobic coverage [53,54,55,56]. However, according to the World Society of Emergency Surgery guidelines [24], quinolones are not recommended due to the high rate of resistance worldwide and should only be used in patients allergic to beta-lactam agents. Carbapenems should always be optimized and used only in very critically ill patients due to the increasing prevalence of carbapenem-resistant Klebsiella pneumoniae [24]. Metronidazole, whose bactericidal spectrum is almost exclusively directed against anaerobes, also shows good penetration into the pancreas [24]. Of the various antibiotics used, only carbapenem prophylaxis shows a trend toward efficacy, although without statistical significance, as shown in the meta-analysis previously analyzed [28]. In light of these observations, this is the class of antibiotics to be used in the first line for patients with suspected infected pancreatic necrosis [26].

## 4. Extrapancreatic Infection

Nonpancreatic infections mainly include pneumonia, bacteremia, central venous catheter infections, urinary tract infections, and cholangitis [37]. In the case of extrapancreatic infection, antibiotic therapy is always recommended [25,26]

## 5. Organism and Resistance Patterns

In SAP, pancreatic or peripancreatic necrosis is initially aseptic and may develop into infected necrosis if contaminated with intestinal bacteria. Translocation of intestinal bacteria is facilitated by intestinal barrier disruption, immunosuppression, and bacterial dysbiosis in combination [3,57] (Figure 1). Most of the causative agents of pancreatic infections are gastrointestinal Gram-negative bacteria resulting from disruption of the intestinal flora and damage to the intestinal mucosa, as well as Gram-positive bacteria, albeit less frequently (such as Staphylococcus aureus, Streptococcus faecalis, Enterococcus), anaerobes, and fungi have been found in infected pancreatic necrosis [24].

A recent study [58] found that intestinal bacteria were the most common source of infection (26.7%) in patients with acute pancreatitis. Laboratory data from blood, urine, sputum, abdominal fluid, catheter, and bile cultures were collected from 2089 patients with AP, and MDR bacteria were determined. Intestinal flora accounted for 26.7% of the total culture results, including in the blood cultures (where Gram-positive bacteria were found in 29.6% of samples) and catheter cultures (where Gram-positive bacteria were found in 20% of samples). The microorganisms with the highest rates of drug resistance were coagulase-positive Staphylococcus aureus (54%), Enterococcus spp. (71%), and Enterococcus faecium (61%). A logistic regression model was used in this study to reveal a link of five parameters with positive cultures (number of antibiotics, length of hospital stay, length of intensive care unit stay, mechanical ventilation, and parenteral nutrition). Furthermore, four parameters (age, hemoglobin, length of hospital stay, and length of antibiotic administration) were found to be positively connected with the distribution of multi-drug-resistant bacterial infections.

## 6. Probiotic Therapy in Acute Pancreatitis

As analyzed earlier, in SAP, pancreatic or peripancreatic necrosis is initially aseptic and may develop into infected necrosis if infected with intestinal bacteria. Several factors contribute to the development of bacterial translocation in AP. Firstly, the increased pancreatic duct pressure and obstruction, characteristic of this condition, may result in the retrograde flow of pancreatic secretions into the common bile duct and duodenum. This retrograde flow can facilitate the migration of bacteria from the gut into the pancreatic tissue. Additionally, the inflammatory response triggered by the release of pancreatic enzymes can cause local tissue damage and compromise the integrity of the intestinal barrier. This disruption allows gut bacteria, including both commensal and potentially pathogenic organisms, to translocate into the systemic circulation and reach distant organs. Moreover, the release of inflammatory mediators, such as interleukins and TNF-α, can impair intestinal barrier function by disrupting tight junctions and reducing the production of protective mucus. The translocated bacteria can elicit an immune response, leading to the release of pro-inflammatory cytokines, the activation of immune cells, and the amplification of the systemic inflammatory cascade [3,4,5,6,7,8,9,10,11,12,13,14,15,16,17,18,19,20,21,22,23,24,25,26,27,28,29,30,31,32,33,34,35,36,37,38,39,40,41,42,43,44,45,46,47,48,49,50,51,52,53,54,55,56,57,58,59].

In this context, the gut microbiome is potentially a promising target for AP treatment because it undergoes disruption during the disease’s etiology. Several studies have identified a dysbiosis of the gut microbiota during AP [60,61,62,63], which consists of an apparent reduction in bacterial diversity in the gut [60], associated with a marked overgrowth of Proteobacteria and Bacteroidetes phyla and a decrease in the abundance of Actinobacteria and Firmicutes compared to healthy controls. At the genus level, an alteration of the intestinal microbiome has been observed in patients with AP, characterized by a decrease in Bifidobacterium and an increase in Enterobacteriaceae and Enterococcus [62]. In summary, the abundance of harmful bacteria, such as Enterococcus, Escherichia coli/Shigella, and several unknown genera from the Enterobacteriaceae family, was significantly higher in patients with AP; in contrast, the abundance of bacteria producing short-chain fatty acids (SCFAs), which are thought to have anti-inflammatory properties, such as several members of the Lachnospiraceae and Ruminococcaceae families, was significantly decreased. Increased intestinal permeability could allow intestinal bacteria to pass through the damaged intestinal barrier in a process known as bacterial translocation, aggravating systemic inflammation and resulting in secondary infections [62]. Probiotics are defined as live microorganisms that help the host’s health when given in sufficient amounts. [64]. The results of studies conducted in recent decades to investigate the efficacy of probiotics in acute pancreatitis have yielded conflicting results. Studies conducted in animal models have provided encouraging results [65], showing a beneficial effect of probiotic administration in experimental models of AP. Unfortunately, randomized clinical trials [66,67] conducted in patients have not confirmed these results. Besselink et al. conducted a multicenter randomized, double-blind, placebo-controlled trial (also named PROPATRIA, Probiotics in Pancreatitis Trial), including 298 patients randomized to receive either a probiotic combination or placebo; surprisingly, results showed an increase in mortality in the group of patients treated with probiotics compared to the control group (mortality: 16% vs. 6%; intestinal ischemia: 6% vs. 0%). [66] Lately, the same authors showed that the effect of probiotic administration was related to the presence of organ failure. Probiotic prophylaxis was linked to an increase in enterocyte damage and an increase in bacterial translocation in patients with AP and concurrent organ failure; in patients without organ failure, prophylaxis with this particular strain combination had no effect on enterocyte damage but reduced bacterial translocation [68]. However, a recently published meta-analysis [69] found that, in Chinese SAP cohorts, the supplemental use of prebiotics, probiotics, and synbiotics was effective in reducing the length of hospital stay, suggesting that their addition to the standard enteral diet could be a potential option for the treatment of SAP patients. In summary, the results of studies on probiotic prophylaxis in patients with AP are inconclusive regarding benefit or risk profile. To date, the use of probiotics is not recommended in severe AP [26].

## 7. Conclusions

According to the latest guidelines, antimicrobials are indicated in AP only when infection is either confirmed or strongly suspected. In recent decades, several RCTs and meta-analyses have been performed to evaluate the efficacy of antibiotic prophylaxis in reducing mortality and incidence of pancreatic and other organ infections in patients with SAP. Unfortunately, the results are inconclusive, likely due to differences in the class of antibiotics administered, timing of therapy, treatment outcomes, and study design. For these reasons, no conclusive judgments can be made about the overall efficacy of prophylactic antibiotics in acute pancreatitis. Future RCTs are urgently required to determine which subgroups may benefit from prophylactic antibiotics. Several studies have shown that antibiotic prophylaxis should be administered early, within 72 h of symptom onset or 48 h of hospitalization, underscoring the importance of timing of antibiotic initiation. The monitoring of blood levels of PCT and CRP can be useful in predicting the likelihood of developing IPN and can help identify patients who will benefit from early antibiotic therapy. Antibiotic prophylaxis is also correlated with risks, including increased prevalence of multi-drug-resistant bacteria and incidence of fungal infection; antimicrobial stewardship is needed in order to optimize the application of antibiotics to obtain the best therapeutic outcomes while avoiding side effects and decreasing the selective pressure that cause resistance emergence. Antibiotics play a critical role in the treatment of IPN and its associated complications. Early initiation of antibiotic therapy has been shown to improve treatment outcomes by reducing infection-related morbidity and mortality. Typically, broad-spectrum antibiotics, which are known to penetrate pancreatic necrosis, are used to cover both Gram-positive and Gram-negative pathogens, with adjustments based on local resistance patterns and microbiologic data. Nevertheless, alternative strategies such as probiotics may offer potential benefits. The gut microbiome is likely a potential target for treatment of AP, as dysbiosis and bacterial translocation play a pivotal role in the pathogenesis of AP and in the development of pancreatic and peri-pancreatic infections. Several studies have identified a dysbiosis of the gut microbiota during AP, characterized by an increase in the abundance of harmful bacteria (such as Enterococcus, Escherichia coli/Shigella, and several unknown genera from the Enterobacteriaceae family) and a decrease in the abundance of bacteria producing short-chain fatty acids (SCFAs), which are thought to have anti-inflammatory properties (such as several members of the Lachnospiraceae and Ruminococcaceae families). Because the evidence on the modulation of the gut microbiota by probiotics remains controversial to date, the use of probiotics is not recommended in severe AP.

## Figures and Tables

**Figure 1 antibiotics-12-01120-f001:**
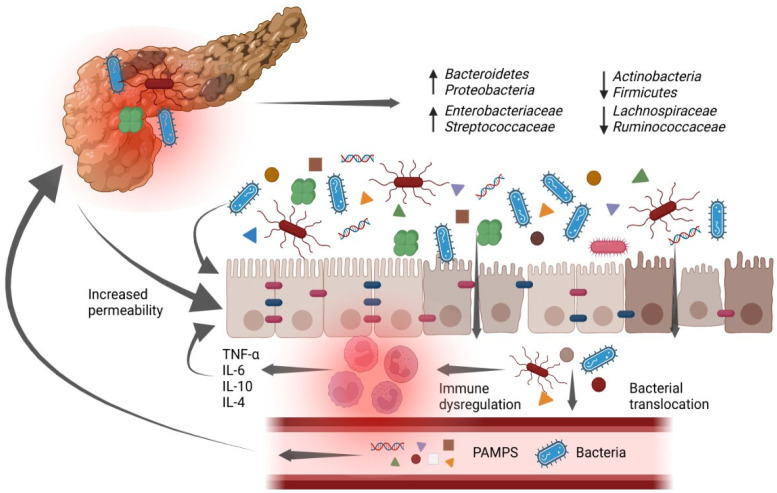
Bacterial translocation as a source of infection of pancreatic necrosis. Dysbiosis and intestinal barrier disruption cause intestinal bacteria to invade through the impaired intestinal barrier in a process known as bacterial translocation. This process exacerbates systemic inflammation and causes secondary infection. The abundance of harmful bacteria is significantly higher in AP patients, whereas the abundance of bacteria that produce short-chain fatty acids, which are thought to have anti-inflammatory properties, is significantly lower. PAMPS: Pathogen-associated molecular patterns. ↑: increase in bacterial abundance; ↓ decrease in bacterial abundance. Created with BioRen-der.com.

**Table 1 antibiotics-12-01120-t001:** Metanalyses on prophylactic antibiotic therapy and main outcomes.

Study	Characteristic	Mortality	Incidence of IPN	Extrapancretic Infection
1998—Golub R. et al. [9]	8 RCTs	⇓	⇓	⇓
2001—Sharma VK et al. [10]	3 RCTs	⇓	No	⇓
2003—Villatoro E et al. [11]	4 RCTs	⇓	⇓	No
2008—Dambrauskas Z et al. [12]	10 RCTs	⇓	⇓	⇓
2008— Bai Y et al. [19]	7 RCTs (467 patients)	No	No	Not investigated
2008—Xu T et al. [29]	8 RCTs (540 patients)	No	⇓	⇓
2009—Jafri NS et al. [20]	8 RCTs (502 patients)	No	No	⇓
2010—Villatoro E et al. [18,30]	7 RCTs (404 patients)	No	No	No
2015—Ukai T et al. [30]	6 RCTs (397 patients)	⇓	⇓	No
2020—Ding N et al. [21]	11 RCTs (747 patients)	No	No	⇓
2022—Guo D et al. [28]	5 RCTs and 2 retrospective observational studies (3864 patients). Only carbapenem antibiotics	No	No	⇓
2022—Poropat G et al. [23]	21 RCTs (1383 patients)	No	No	Statistically significant reduction in the incidence of—infections but no significant difference across the subgroup analysis

⇓: statistically significant reduction.

## Data Availability

Not applicable.

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
