# Peer review of "Antibiotic Utilization in Acute Pancreatitis: A Narrative Review"

_antibiotics, 2023, doi:10.3390/antibiotics12071120_

Round 1
Reviewer 1 Report
The authors have put in a systematic review on the recent developments in the antibiotics treatment in acute pancreatitis. The coverage of the essential points in the management are adequate and informative. The compilation of the review report is lucid and systematic. I will suggest the authors to also include few lines about the step up approach of treatment of IPN in the section on "Infected pancreatitis and antibiotic therapy" along with the antibiotics treatment.
Author Response
Rome, 23 June 2023
Dear Reviewer,
We would like to thank you for the careful assessment of our paper and for your precious comments, that improved the quality of our paper.
We have done our best to address comments satisfactorily and hope that you will appreciate the revised version of the paper. Please find below a point-by-point response to your comments.
Best regards
Enrico Celestino Nista, on behalf of all co-authors
POINT-BY-POINT RESPONSE
I will suggest the authors to also include few lines about the step-up approach of treatment of IPN in the section on "Infected pancreatitis and antibiotic therapy" along with the antibiotics treatment.
R: The text has been implemented as requested
Reviewer 2 Report
The manuscript “Antibiotic utilization in Acute Pancreatits: A Narrative Review” discusses the use of antibiotics in acute pancreatitis prophylaxis and treatment. Similar to all other use of antibiotics, overgenerous use in pancreatitis leads to multidrug resistance. It is also important to consider the potential benefit to the patient as well as the potential side effects of antibiotics treatment, such as fungal infections.
The manuscript thoroughly discusses antibiotic uses as well as probiotics in the treatment of acute pancreatitis and this subject is of general interest in the context of increasing multidrug resistant infections. The manuscript is generally well written, but there are some minor concerns:
1: Overall, the manuscript would benefit from a reduction in the number of acronyms used. Why does bacterial translocation need an acronym? Have you explained what RCT stands for?
Line 148: What is SIRS?
2: Please refrain from using one sentence as a paragraph.
3: Line 175: Please refrain from using contracted form. Write “did not” instead of didn´t.
4: Line 345: What does “PCT e CRP serum measurements” mean? This is not English.
The language is good, no major problems. One contracted form detected and one Italian word (PCT e CRP) in line 345.
Author Response
Rome, 23 June 2023
Dear Reviewer,
We would like to thank you for the careful assessment of our paper and for your precious comments, that improved the quality of our paper.
We have done our best to address comments satisfactorily and hope that you will appreciate the revised version of the paper. Please find below a point-by-point response to your comments.
Best regards
Enrico Celestino Nista, on behalf of all co-authors
POINT-BY-POINT RESPONSE
Overall, the manuscript would benefit from a reduction in the number of acronyms used. Why does bacterial translocation need an acronym? Have you explained what RCT stands for?
R: The meaning of “RCT” has been explained. The acronym for bacterial translocation has been removed.
Line 148: What is SIRS?
R: The meaning of “SIRS” has been explained.
Please refrain from using one sentence as a paragraph.
R: The text has been modified as requested.
Line 175: Please refrain from using contracted form. Write “did not” instead of didn´t.
R: The text has been modified as requested.
Line 345: What does “PCT e CRP serum measurements” mean? This is not English.
R: The text has been rephrased to improve grammar